# Ultrafast growth of nanocrystalline graphene films by quenching and grain-size-dependent strength and bandgap opening

Tong Zhao[1,2,6], Chuan Xu[1,6], Wei Ma [1,2,6], Zhibo Liu[1], Tianya Zhou[1,2], Zhen Liu[3], Shun Feng[1], Mengjian Zhu[4], Ning Kang[3], Dong-Ming Sun [1], Hui-Ming Cheng [1,2,5] & Wencai Ren [1,2]*

Nanocrystallization is a well-known strategy to dramatically tune the properties of materials; however, the grain-size effect of graphene at the nanometer scale remains unknown experimentally because of the lack of nanocrystalline samples. Here we report an ultrafast growth of graphene films within a few seconds by quenching a hot metal foil in liquid carbon source. Using Pt foil and ethanol as examples, four kinds of nanocrystalline graphene films with average grain size of ~3.6, 5.8, 8.0, and 10.3 nm are synthesized. It is found that the effect of grain boundary becomes more pronounced at the nanometer scale. In comparison with pristine graphene, the 3.6 nm-grained film retains high strength (101 GPa) and Young's modulus (576 GPa), whereas the electrical conductivity is declined by over 100 times, showing semiconducting behavior with a bandgap of ~50 meV. This liquid-phase precursor quenching method opens possibilities for ultrafast synthesis of typical graphene materials and other two-dimensional nanocrystalline materials.

[1] Shenyang National Laboratory for Materials Science, Institute of Metal Research, Chinese Academy of Sciences, Shenyang 110016, China. [2] School of Materials Science and Engineering, University of Science and Technology of China, Shenyang 110016, China. [3] Key Laboratory for the Physics and Chemistry of Nanodevices and Department of Electronics, Peking University, Beijing 100871, China. [4] College of Advanced Interdisciplinary Studies, National University of Defense Technology, Changsha 410073, China. [5] Shenzhen Geim Graphene Center, Tsinghua-Berkeley Shenzhen Institute (TBSI), Tsinghua University, Shenzhen 518055, China. [6] These authors contributed equally: Tong Zhao, Chuan Xu, Wei Ma. *email: wcren@imr.ac.cn

Grain boundary (GB) is a common structural feature in mass-produced large-size materials and, more importantly, it provides an efficient way to dramatically tune the properties of materials. In bulk nanocrystalline materials, for instance, the collective effect of abundant GBs leads to many unusual properties that are absent in their monocrystalline, even microcrystalline, counterparts such as superplastic extensibility of nanocrystalline copper[1], ultralow magnetocrystalline anisotropy of nanocrystalline soft magnets materials[2], and high photovoltaic efficiency of nanocrystalline titanium oxide[3]. Pristine graphene is well known as a super material possessing giant carrier mobility, record thermal conductivity, and extremely high mechanical strength[4]. The GB in graphene is a topological line defect formed by chains of pentagons, heptagons, and distorted hexagons[5]. Theoretical and experimental studies show that individual graphene GBs have distinct electronic, magnetic, thermal, and mechanical properties that strongly depend on their atomic arrangement[6–13]. Similar to bulk materials, however, the graphene films produced by scalable method contain abundant GBs with random atomic arrangements and orientations[5–8,14,15]. Therefore, in addition to the studies of individual GBs, understanding the collective effect of GBs on the properties of graphene films, in particular for nanocrystalline graphene (NG) films, is essentially important not only for fundamental studies but also for technological applications.

However, it remains an open challenge to produce monolayer NG films by the existing synthesis methods such as chemical vapor deposition (CVD). According to the Arrhenius equation, a very high concentration of active carbon species is required to achieve a high nucleation density[16], the prerequisite to obtain NG, which inevitably leads to the formation of multilayers. A low growth temperature is beneficial for achieving high nucleation density due to the decreased desorption rate of active carbon species[16], but on the other hand it is also unfavorable for nucleation, because only low concentration of active carbon species can be produced. So far, the monolayer-dominated graphene films produced by traditional CVD usually have an average grain size over hundreds of nanometers[5–8,15]. Recently, it has been demonstrated that NG could be transformed from a thin layer of solid carbon source, such as aromatic self-assembled monolayers and photoresist film, on dielectric or metal substrates by high-temperature graphitization[17–19]. However, the products are dominantly non-uniform multilayers with many structure defects and large portion of incompletely graphitized or $sp^3$ domains. Owing to the difficulty in the synthesis of NG films, the collective effect of GBs at the nanometer scale still remains unknown experimentally so far.

Here we report a method for ultrafast synthesis of graphene films within a few seconds by quenching a high-temperature metal foil in liquid carbon source. Using Pt foil and ethanol as examples, it is found that the grain size of graphene films decreases with decreasing the onset temperature of the metal foil, which enables the controlled synthesis of NG films with different average grain size of ~3.6, 5.8, 8.0, and 10.3 nm. Using these materials, we reveal the effect of grain size on the mechanical and electrical properties of graphene at the nanometer scale and determine the scaling laws. We also demonstrate the ultrafast quenching synthesis of bilayer-dominated NG films and high-quality multilayer graphene foams by changing the substrate.

## Results

**Quenching growth of NG films**. To grow NG films by liquid carbon source quenching, as an example, we used Pt foil with high catalytic activity as the substrate and ethanol with low toxicity as the liquid carbon source. The quenching process is illustrated in Fig. 1a and is shown in Supplementary Movie 1. After polishing and cleaning, the Pt foil was heated to a high temperature in argon atmosphere and then rapidly quenched in liquid ethanol at room temperature. On the one hand, ethanol provides an all carbon surrounded environment for Pt foil. At the very beginning of quenching, the high-temperature Pt foil leads to

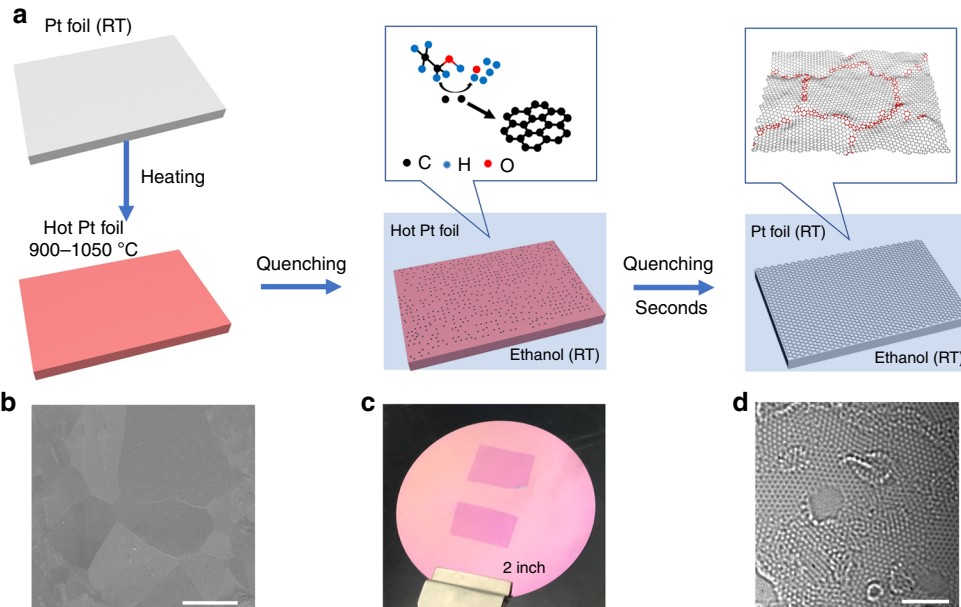

**Fig. 1** Ultrafast synthesis of NG film by ethanol quenching. **a** Schematic of the synthesis process of NG film by quenching a hot Pt foil in ethanol at room temperature (RT). **b** Scanning electron microscopy (SEM) image of an as-grown NG film on Pt substrate. The different contrasts represent the Pt grains with different orientations. **c** Two pieces of NG films (~1 × 1.5 cm²) grown on Pt foils with onset temperature of 1050 °C (top) and 900 °C (bottom), and then transferred onto SiO₂/Si substrate. **d** Atomic-level HRTEM image of the graphene film grown with an onset temperature of 900 °C, showing nanocrystalline structure. The strong adsorption ability of GBs leads to PMMA residues[5]. To expose the GBs, the samples were annealed at 320 °C in air for 8 h to remove the adsorbed PMMA residues, which simultaneously generates some holes. Scale bars: **b** 100 μm; **d** 2 nm

rapid decomposition of ethanol to enable an extremely high nucleation density. On the other hand, ethanol acts as a cooling medium to provide a non-equilibrium growth environment. The rapidly cooled Pt foil not only significantly reduces the desorption rate of active carbon species[16] but also greatly suppresses the decomposition of ethanol. The negligible carbon desorption remains the original high nucleation density, whereas the reduced carbon feeding ensures the growth and stitching of graphene nucleus and avoids the formation of multilayers. As a result, after quenching for a few seconds, a continuous and uniform mono-layer NG film was synthesized (Fig. 1b–d).

For structural characterizations and property measurements, we transferred the NG films from Pt foils onto SiO$_2$/Si substrates and transmission electron microscope (TEM) grids by the electrochemical bubbling method[14] with poly(methyl methacry-late) (PMMA) as a support layer. Figure 1c shows a photograph of two NG films (~1 × 1.5 cm$^2$) grown with onset temperature of 1050 °C and 900 °C and then transferred onto a 2-inch SiO$_2$/Si substrate. Figure 1d is a typical high-resolution TEM (HRTEM) image of the NG film grown with onset temperature of 900 °C, showing that the grain size is around 3 nm. X-ray photoelectron spectroscopy (XPS) measurements show that the NG films are free of oxidation even though oxygen-containing carbon source (ethanol) was used (Supplementary Fig. 1). In principle, there is no size limit for the film and large NG films can be synthesized by simply using large Pt foils. In addition, the electrochemical bubbling transfer method allows for the repeated use of the Pt substrate (Supplementary Fig. 2). More importantly, the growth rate of NG films by ethanol quenching is extremely fast. It takes only a few seconds to achieve a completely continuous film (Fig. 1b, c and Supplementary Movie 1). In contrast, over tens of minutes are usually required to grow a graphene film for traditional CVD[5–8,14,15].

The synthesis of NG film by ethanol quenching strongly depends on the onset temperature of the Pt foil, which affects the quenching rate and consequently the carbon feeding. We simulated the quenching of a Pt foil with different onset temperatures in ethanol at room temperature by COMSOL, which has been widely used to simulate the quenching process of metals in various mediums, and the results are shown in Supplementary Fig. 3 and Supplementary Table 1. It can be found that the real onset temperature of Pt foil during ethanol quenching is a bit lower than the initial onset temperature because of the cooling in air before immersing into ethanol. More importantly, the quenching time, during which the temperature decreases from the real onset temperature to the decomposition temperature of ethanol, decreases with decreasing the onset temperature, leading to a decrease in carbon feeding. As a result, only discontinuous NG films were synthesized with low onset temperatures of 850 °C and 800 °C because of insufficient carbon supply (Supplementary Fig. 4a, b), whereas a great number of adlayers were formed on the surface of monolayer graphene film with a high onset temperature of 1100 °C (Supplementary Fig. 4c), which is a typical characteristic of graphene growth in the case of excessive carbon supply. Therefore, we synthesized NG films on Pt foils with onset temperatures in the range of 900–1050 °C. As shown below, the grain size of NG films synthesized in such temperature range decreases with decreasing the onset temperature.

**Structural characterizations of NG films**. We first carried out detailed characterizations on the graphene films grown with an onset temperature of 1050 °C. As shown in Fig. 2a, the film has a very uniform thickness. Atomic force microscopy (AFM) and UV–vis-NIR spectroscopy measurements indicate that the film

has a thickness of ~0.72 nm (Fig. 2b) and optical transmittance of 97.3% at wavelength of 550 nm (Fig. 2c), confirming that it is monolayer[20]. Significantly, different from the pristine graphene and polycrystalline graphene, even those with a grain size of ~220 nm, our film shows prominent defect-related Raman $D$ and $D'$ peaks ($I_D/I_G$ ~ 1.5, $I_D/I_G$ ~ 0.4; Fig. 2d). It has been reported[21] that the intensity ratio of $D$ to $D'$ peak, $I_D/I_{D'}$, does not depend on the defect concentration, but only on the type of defect. It is ~13 for defects associated with $sp^3$ hybridization, ~7 for vacancy-like defects, and ~3.5 for GB-like defects[21]. It is noteworthy that the Raman spectra taken from different positions in our sample show almost the same $I_D/I_{D'}$ of ~3.5, suggesting that the defects are GBs. According to the equation $L_g = 560/E^4_{laser}(I_D/I_G)^{-1}$, which is widely used to evaluate the grain size of graphene and gra-phite[22], the average grain size ($L_g$) of our sample was estimated to be ~12 nm.

In order to obtain the atomic-level structure information of the graphene film, we performed aberration-corrected HRTEM measurements. A low electron beam voltage of 80 kV was used to minimize the irradiation damage to the graphene. The electron diffraction (ED) ring pattern confirms the nanocrystalline structure of our graphene film (inset of Fig. 2e). Because of the stronger adsorption ability of GBs than the perfect graphene lattice[5], the PMMA residues tend to adsorb on the GBs to form a network, which separates the film into many domains with very clean surface (Fig. 2e). Further observations show that the neighboring domains have different crystal orientations and very high crystallinity free of defects (Fig. 2e, f). We then obtained the grain-size distribution by measuring 111 grains. As shown in Fig. 2g, the grain sizes are distributed in the range of 5–17 nm with an average of ~10.3 nm, consistent with the Raman results.

Figure 3 shows the NG films synthesized with onset temperatures of 1000 °C and 950 °C. The sample synthesized with an onset temperature of 1000 °C still shows well-defined Raman peaks, but $I_D/I_G$ is increased and $I_{2D}/I_G$ is decreased compared with those of the 10.3 nm-grained NG films (Fig. 3a), a characteristic of the decrease in grain size[23]. Importantly, when further decreasing the onset temperature to 950 °C, both the $D$ and $G$ peaks are broadened together with the merging of $G$ and $D'$ peaks, and the 2D peak is significantly decreased (Fig. 3b). As reported previously[23], if the grain size is smaller than the coherence length of optical phonons (~30 nm), the Raman-allowed phonon wavevector $q$ is relaxed due to the spatial confinement of the crystallites, leading to the broadening of the Raman peaks. Moreover, the width of both $G$ peak and $D$ peak increases exponentially as grain size decreases[23,24]. Therefore, the above Raman features suggest the further decrease in the grain size. Aberration-corrected HRTEM images confirm that both samples are NG possessing high crystallinity within the grains (Fig. 3c, d). Grains (102 and 120) were further measured for the samples synthesized with onset temperature of 1000 °C and 950 °C, respectively. The results show that their grain sizes are distributed in the range of 2–16 and 2–14 nm, respectively, with an average of ~8.0 and ~5.8 nm (Fig. 3e, f).

Importantly, we were able to synthesize high-quality NG films with average grain size below 5 nm by further decreasing the onset temperature of Pt foils to 900 °C (Fig. 4). The uniform optical contrast, ~0.74 nm thickness, and ~97.1% optical transmittance indicate that the film is of uniform monolayer (Fig. 4a–c). It is worth noting that the film shows significantly different Raman features from those of NG films grown with onset temperatures of 950 °C, 1000 °C, and 1050 °C (Fig. 4d). Both the $G$ and $D$ peaks become much broader, $I_D/I_G$ is reduced to 1.3, and the 2D peak almost disappears. When the grain size is larger than ~6 nm, the average width of GBs plus the coherence length of electrons/holes, $I_D/I_G$ will keep increasing with

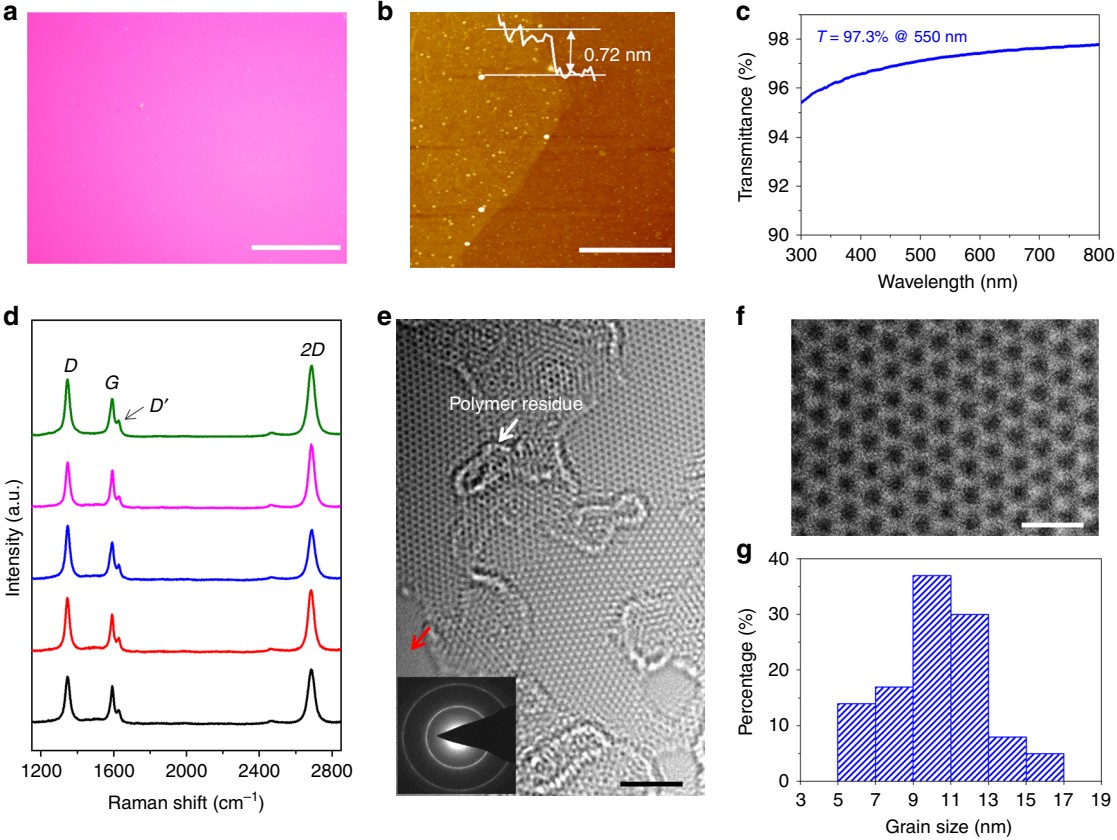

**Fig. 2** Characterizations of the 10.3 nm-grained NG film. **a** Optical image. **b** AFM image. The white dots are contaminations left after transfer. **c** Optical transmittance spectrum measured by UV–vis–NIR spectrometer. **d** Raman spectra taken from randomly selected five positions in **a**. **e** Typical HRTEM image. Inset, ED pattern. To expose the GBs, the samples were annealed at 320 °C in air for 8 h to remove the adsorbed PMMA residues, which simultaneously generates some holes as denoted by a red arrow. **f** Atomic structure of the interior of a grain. **g** The histogram of the grain size extracted from HRTEM measurements. Scale bars: **a** 50 μm; **b** 500 nm; **e** 2 nm; **f** 0.5 nm

decreasing grain size as shown above[23]. However, when the grain size is less than ~6 nm, the neighboring GBs are closer than the average distance an electron-hole pair travels before scattering with a phonon. In this case, the contributions of GBs to the $D$ peak intensity will not sum independently and $I_D/I_G$ will decrease with decreasing grain size[23,25]. Therefore, the reduced $I_D/I_G$ and the disappearance of $2D$ peak suggest a grain size smaller than ~6 nm and a great change in the electronic structure[23,25]. We further used aberration-corrected HRTEM to characterize the atomic-level structure of the film. Similar to the 10.3, 8.0, and 5.8 nm-grained films, the interior of the grains have very high crystallinity with very few PMMA residues (Fig. 4e, f), which allows identification of their lattice orientations. The exposed GB shows that the neighboring domains are well-stitched together without overlapping and buckling (Fig. 4f). We further obtained the grain-size distributions by measuring 108 grains. As shown in Fig. 4g, the grain size is distributed in the range of ~1–9 nm with an average of ~3.6 nm.

**Mechanical properties of NG films**. The above NG films provide an ideal platform to investigate the effect of grain size on the properties of graphene at the nanometer scale, a blank area that has not been revealed experimentally. We first measured the mechanical properties of NG films by using AFM nanoindentation (Fig. 5a) as reported previously[8,26]. To do this, the NG films were transferred onto a SiO2/Si substrate with an array of holes with 1.2 μm diameter and 300 nm depth to create suspended films. To avoid the indenter tip radius effect of a small diamond

tip, AFM indentation on the suspended NG film was performed by a large diamond tip with a radius of 61.58 nm and spring constant of 48.46 N m$^{-1}$, to obtain force ($F$) vs. displacement ($\delta$) curves (Fig. 5b), which can be approximated as

$$F = \left(\sigma_0^{2D}\pi\right)\delta + \left(\frac{E^{2D}q^3}{a^2}\right)\delta^3 \qquad (1)$$

where $\sigma_0^{2D}$ is the prestress, $E^{2D} = E \cdot h$ is the 2D Young's modulus with a thickness of $h$ where $E$ is the Young's modulus; $q = 1/(1.05 - 0.15\nu - 0.16\nu^2)$ is a constant where $\nu$ is the Poisson's ratio of the film and $a$ is the radius of the hole. In our measurements of NG films, we took $h = 0.335$ nm, $q = 1.02$, and $a = 0.6$ μm. We used a linear model to evaluate the 2D fracture strength $\sigma^{2D}$ by

$$\sigma^{2D} = \sqrt{F_{\max}E^{2D}/4\pi r_{\mathrm{tip}}} \qquad (2)$$

where $F_{\max}$ is the corresponding fracture force and $r_{\mathrm{tip}}$ is the radius of the tip. Facture strength $\sigma = \sigma^{2D}/h$. For the 10.3, 8.0, 5.8, and 3.6 nm-grained NG films, the extracted mean Young's modulus is 892, 776, 609, and 576 GPa, respectively (Fig. 5c), and the mean fracture strength is 127, 108, 104, and 101 GPa (Fig. 5d). For comparison, we also used the same method to measure the mechanical properties of single-crystal graphene domains (~1 mm) and the 220 nm-grained polycrystalline graphene films[14,15] (Supplementary Fig. 5). Compared with the single-crystal graphene, it is worth noting that the Young's modulus and fracture strength are decreased by 9%, 17%, 28%, 43%, and 46%, and 9%, 25%, 36%, 39%, and 40%, respectively, for the

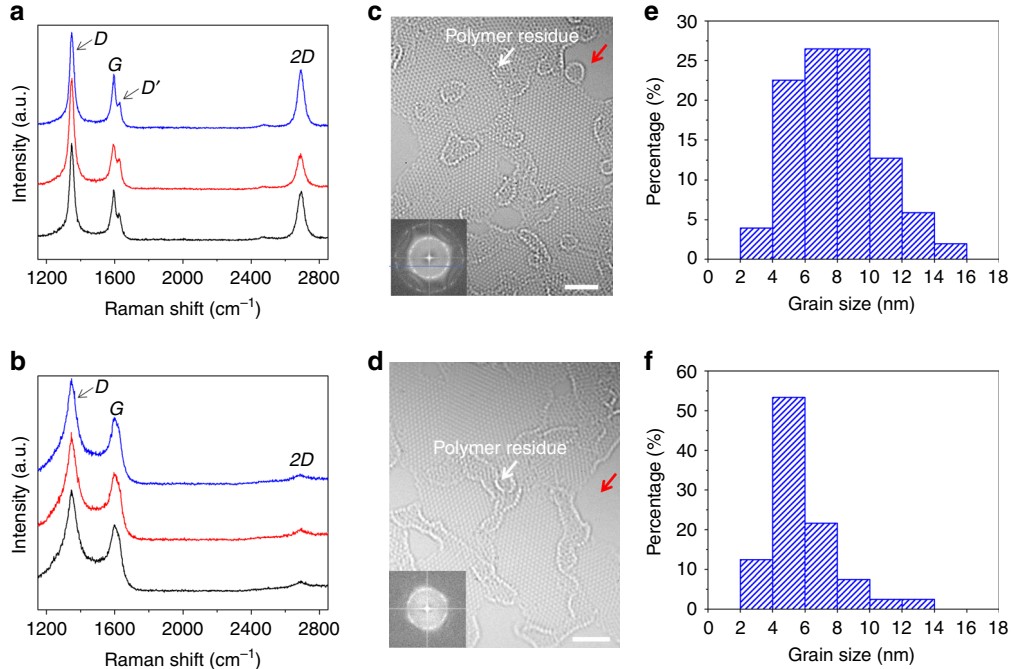

**Fig. 3** Characterizations of the 8.0 nm-grained and 5.8 nm-grained NG films. **a, c, e**, Raman spectra taken from randomly selected three positions **a**, typical HRTEM image **c**, and the histogram of the grain size extracted from HRTEM measurements **e** of 8.0 nm-grained NG film. **b, d, f**, Raman spectra taken from randomly selected three positions **b**, typical HRTEM image **d**, and the histogram of the grain size extracted from HRTEM measurements **f** of 5.8 nm-grained NG film. Insets in **b** and **d**, Fourier transform patterns. To expose the GBs, the samples were annealed at 320 °C in air for 8 h to remove the adsorbed PMMA residues, which simultaneously generates some holes as denoted by a red arrow. Scale bars: **c, d**, 2 nm

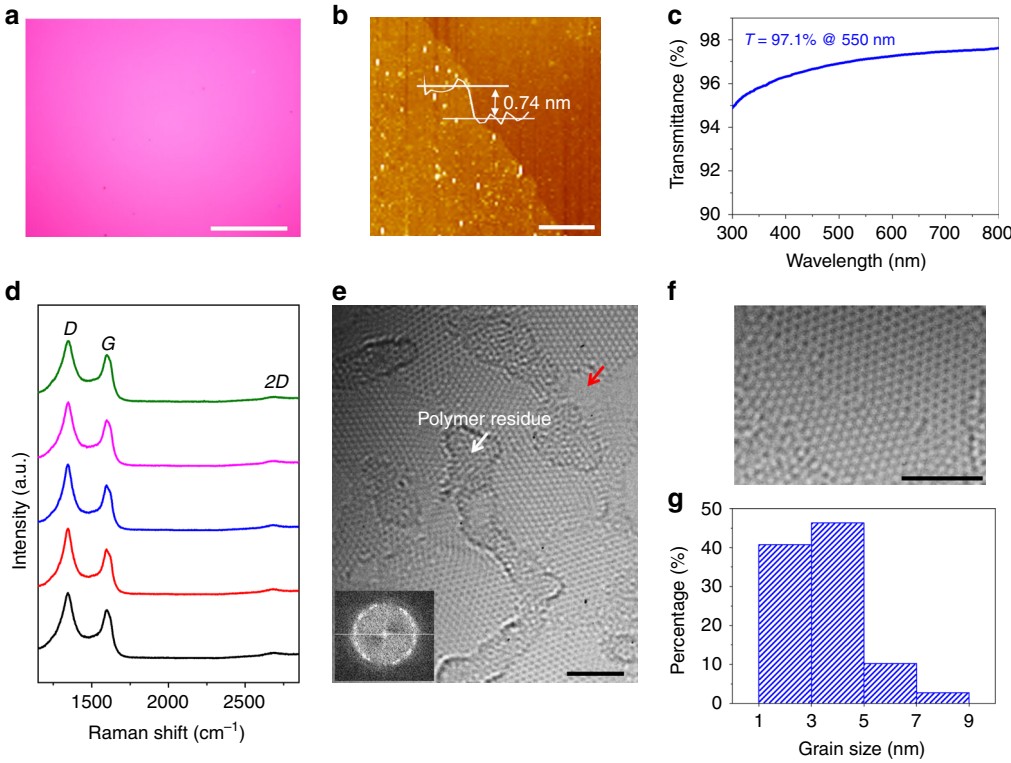

**Fig. 4** Characterizations of the 3.6 nm-grained NG film. **a** Optical image. **b** AFM image. The white dots are the contaminations left after transfer. **c** Optical transmittance spectrum measured by UV–vis–NIR spectrometer. **d** Raman spectra taken from randomly selected five positions in **a**. **e** Typical HRTEM image. Inset, Fourier transform pattern. To expose the GBs, the samples were annealed at 320 °C in air for 8 h to remove the adsorbed PMMA residues, which simultaneously generates some holes as denoted by red arrow. **f** Atomic structure of a region containing a GB. **g** The histogram of the grain size extracted from HRTEM measurements. Scale bars: **a** 50 μm; **b** 2 μm; **e, f** 2 nm

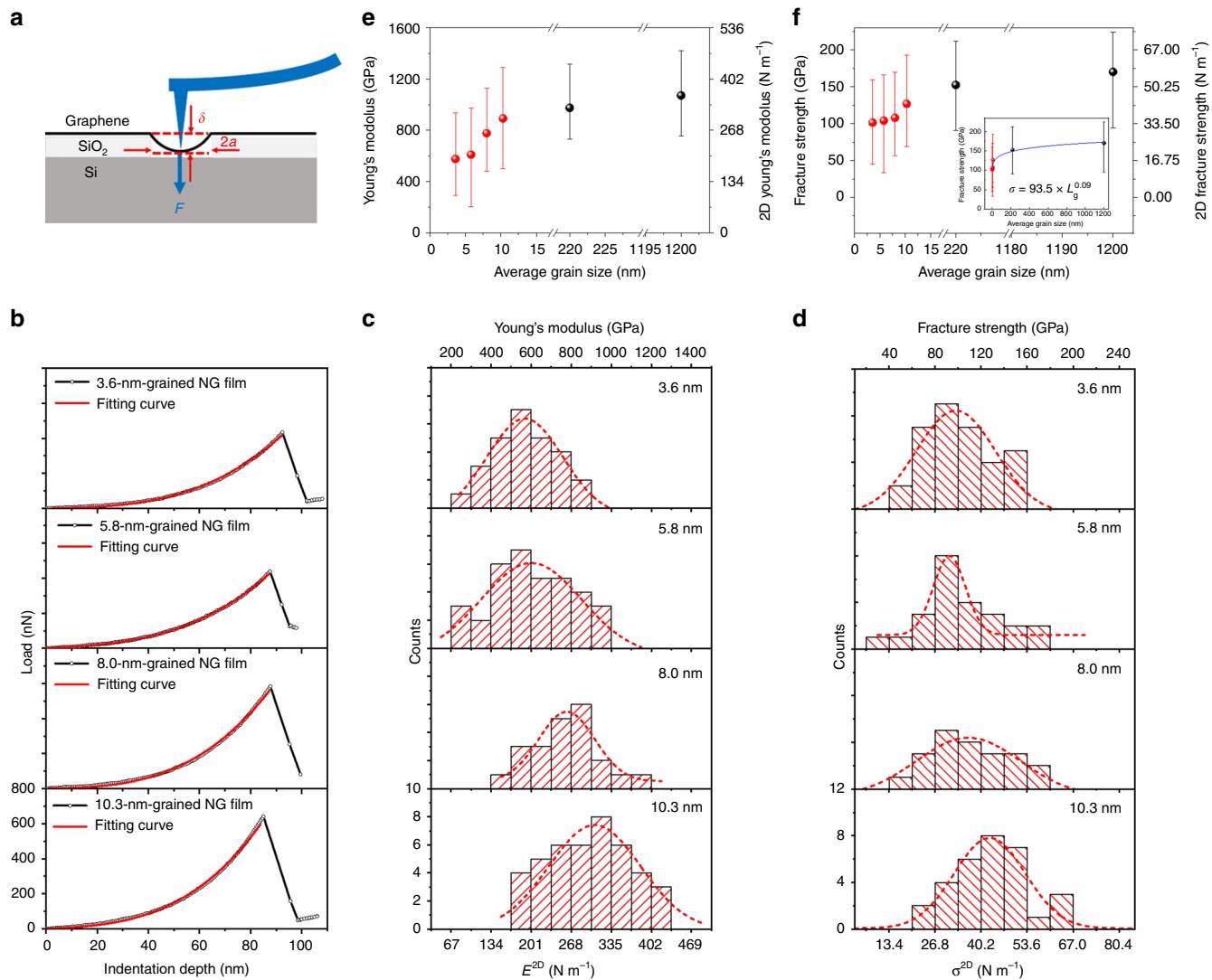

**Fig. 5** Mechanical properties of NG films. **a** Schematic of nanoindentation on a suspended NG film. **b** Representative force-displacement curves of the 3.6, 5.8, 8.0, and 10.3 nm-grained NG films in AFM nanoindentation. The red lines are fitting curves to Eq. 1. **c**, **d** The histograms of the Young's modulus **c** and fracture strength **d** of 3.6, 5.8, 8.0, and 10.3 nm-grained NG films. The dashed lines indicate the fitted Gaussian distributions. **e**, **f** The Young's modulus **e** and fracture strength **f** as a function of grain size. The blue line in the inset of **f** is a fit to the experimental data, which gives a fitted equation $\sigma = 93.5 \times L_g^{0.09}$. Here, the diameter (1.2 μm) of the holes in mechanical property measurements was used as the grain size of single-crystal graphene domains even though the real gain size is ~1 mm. The error bars in **e** and **f** show the variation of the Young's modulus and fracture strength, respectively

220 nm-grained polycrystalline graphene, 10.3, 8.0, 5.8, and 3.6 nm-grained NG. These results suggest that the mechanical properties of graphene reduce more pronounced with decreasing grain size but still retain high values even at nanometer scale.

The above results give the first experimental evidence of the effect of grain size on the mechanical properties of graphene at the nanometer scale, which exists open controversy in theoretical predictions[27]. Such mechanical behavior of NG films is remarkably different from nanocrystalline metals[28–30]. For nanocrystalline metals, a reduction in grain size only leads to a slight decrease in elastic constant compared with the corresponding polycrystals. As for NG films, GB is a low-elastic modulus component, because the bonds in GBs are usually longer than those in the grains[31,32]. Compared with bulk nanocrystalline metals, NG films show more pronounced decrease in Young's modulus with decreasing grain size (Fig. 5e), indicating more significant effect of GB on the Young's modulus in this atomically thin system. The strength of metals is governed by dislocation

nucleation, motion, and interaction. The motion of dislocation is impeded by the presence of GB. Therefore, nanocrystalline metals are significantly harder than their microcrystalline counterparts, well known as the Hall–Petch effect[28]. However, the GBs in graphene are composed of stable covalent bonds that seldom move under external loading at room temperature. Therefore, the brittle breakage of $sp^2$ bonds, rather than motion of dislocations, dominates the failure behavior of graphene. Previous studies show that the fracture of polycrystalline graphene preferentially starts at the junction of GBs because of the accumulation of prestrain, initiating an intergranular crack, and then the crack kink into the adjoining grains because of the more complex stress state[8,27,33]. The slightly reduced strength of GBs leads to the strength decrease of NG films. Theoretically, such failure can be described by the weakest-link model[33], in which the failure strength follows a power-law relation with the number of GB junctions. The number of GB junctions is roughly scaled with the grain size in an inverse quadratic relation[34]. As a result, the

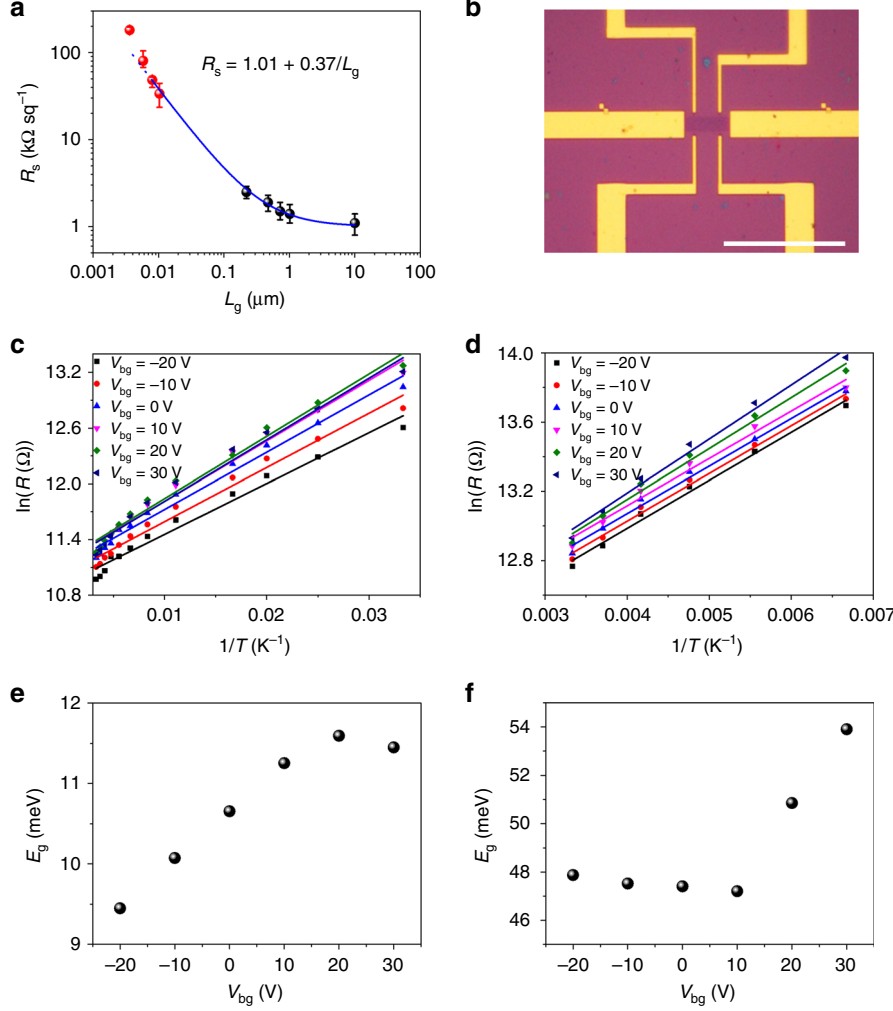

**Fig. 6** Electrical transport properties of NG films. **a** The sheet resistance as a function of grain size. The data of polycrystalline graphene films with average grain size of ~220 nm, 470 nm, 720 nm, 1 μm, and 10 μm was reproduced from ref. [15]. The error bars show the variations of sheet resistance. The blue line is the fitting curve to the equation $R_s = R_{Gs} + \rho_{GB}/L_g$. **b** Optical image of a Hall bar device made with a NG film. **c**, **d** The resistance in ln scale as a function of the inverse of temperature for the 10.3 nm-grained **c** and 3.6 nm-grained **d** NG films at different values of gate voltages. The temperature range is 30–300 K for 10.3 nm-grained NG and 150–300 K for 3.6 nm-grained NG. The lines are the fittings to the equation $R(T) \propto \exp(E_a/kT)$, showing activated behavior. **e**, **f** The extracted bandgaps as a function of gate voltage for the 10.3 nm-grained **e** and 3.6 nm-grained **f** NG films. Scale bar: **b**, 20 μm

fracture strength of NG film follows an inverse pseudo Hall–Petch relation with the grain size, $\sigma = 93.5 \times L_g^{0.09}$ (Fig. 5f).

**Electrical properties of NG films**. We then measured the electrical properties of the NG films using a four-probe station at room temperature. The average sheet resistance obtained is ~33.7, 48.5, 80.3, and ~181.6 kΩ sq$^{-1}$ for the 10.3, 8.0, 5.8, and 3.6 nm-grained NG films, respectively, corresponding to the electrical conductivity of $8.9 \times 10^4$, $6.2 \times 10^4$, $3.7 \times 10^4$, and $1.6 \times 10^4$ S m$^{-1}$. In order to gain a better understanding of the relationship between grain size and electrical properties, we plotted these four data together with those reported[15] for polycrystalline graphene with grain size from 220 nm to 10 μm (Fig. 6a), and fitted them using the equation $R_s = R_{Gs} + \rho_{GB}/L_g$[35], where $R_s$ is the sheet resistance of the graphene film, $R_{Gs}$ is the sheet resistance within the grain, $\rho_{GB}$ is the GB resistivity, and $L_g$ is the grain size. It is noteworthy that the increase in sheet resistance is speeded up dramatically with decreasing the grain size, in particular at the nanometer scale. In comparison with single-crystal graphene, the sheet resistance is increased by

only ~2 times for the 220 nm-grained graphene but over two orders of magnitude for NG. Moreover, it is worth noting that no good fit can be obtained for all the data (Supplementary Fig. 6), but an excellent fit was achieved when the data for 5.8 and 3.6 nm-grained NG were excluded (Fig. 6a). The fitted $R_{Gs} = 1.01$ kΩ sq$^{-1}$ and $\rho_{GB} = 0.37$ kΩ μm are well consistent with those reported previously[15]. The unexpectedly large resistance of 5.8 and 3.6 nm-grained NG indicates strongly enhanced influence of the GBs on the electronic properties. It has been reported that the GB can perturb the carrier scattering over a length scale of the order of 20 nm[36]. We suggest that the strong coupling between neighboring GBs in the 5.8 and 3.6 nm-grained NG is one important reason for the stronger effect of GBs on the charge transport.

We further fabricated Hall bar devices to investigate the electrical transport properties of NG films at different temperatures (Fig. 6b). Because of the strong adsorption ability of GBs[5], the adsorbed PMMA at the GBs results in hole doping in graphene devices (Supplementary Fig. 7). Consistent with the room-temperature four-probe measurements, the

3.6 nm-grained NG shows about five to ten times higher resistance than the 10.3 nm-grained NG at the same measurement conditions. Importantly, both NG samples show increased resistance with decreasing the measuring temperature (Fig. 6c, d), suggesting the opening of a bandgap. This is in sharp contrast to the pristine graphene even polycrystalline graphene, which are well known as good conductors and exhibit metallic behavior in a broad range of temperature[5–7,14,15,37]. Especially, the resistance of the 3.6 nm-grained NG increases steeply with decreasing temperature, suggesting that the bandgap becomes larger with decreasing grain size. We then examined the temperature dependence of the resistance with thermally activated transport behavior at low temperatures, whereby the resistance, $R$, varies temperature as $R\,(T) \propto \exp(E_a/kT)$, where $k$ is Boltzmann's constant and $E_a$ is the activation energy, corresponding to half the bandgap ($E_g$)[38]. We fitted the resistance vs. temperature curves at different fixed gate voltages using the above equation (Fig. 6c, d). As shown in Fig. 6e, f, the extracted bandgap is ~50 meV and ~10 meV for the 3.6 nm-grained NG and 10.3 nm-grained NG, respectively, approximately following $E_g \propto L_g^{-1}$. Such grain size dependence of the bandgap indicates that the bandgap may originate from the quantum confinement and the crucial effect of the GBs, as in the case of graphene nanoribbons[39].

## Discussion

Our work demonstrates the ultrafast synthesis of NG films with grain size below 10 nm by a liquid carbon source quenching method and reveals the grain-size effect on the mechanical and electrical properties of graphene at nanometer scale. Interestingly, the 3.6 nm-grained NG films still retain a high mechanical strength of 101 GPa but show a semiconducting transport behavior totally different from the graphene with large grain size. Moreover, the scaling laws of both mechanical strength and electrical resistance as a function of grain size were derived. These findings not only resolve the controversies in theoretical predictions on grain-size effect but also provide guidelines to tailor the properties of graphene by grain-size engineering.

Importantly, for the liquid carbon source quenching method, there is plenty of room for improvement. Further optimization of the growth process, including substrate, onset temperature, quenching rate, and liquid carbon source, may lead to ultrafast controlled synthesis of NG films and other graphene materials beyond NG. For instance, the use of a thicker Pt foil (1 mm) significantly reduces the quenching rate because of the storage of more heat (Supplementary Fig. 8), which allows a longer time for ethanol decomposition. The increased carbon feeding results in the synthesis of bilayer-dominated NG films (Supplementary Fig. 9). Moreover, the grain size of NG films can also be tuned by the thickness of Pt foil. Even at a high onset temperature of 1050 °C, NG films with smaller grain size can be synthesized by using thinner Pt substrate because of the less storage of heat (Supplementary Fig. 10). We have also been able to synthesize high-quality three-dimensional multilayer graphene foam within a few seconds by using nickel foam as a substrate (Supplementary Fig. 11 and Movie 2), which is about two orders of magnitude faster than the traditional CVD method[40].

Besides ultrafast growth rate, this quenching method also has outstanding advantages over the traditional CVD methods in terms of energy-saving, ease of operation, and scalability, which guarantee its great potential for industrial production of graphene materials. In principle, this liquid-phase precursor quenching method also can be used to synthesize other 2D materials with nanometer-sized grains. Such materials will open up the

possibilities for investigating the predicted unique properties and applications of nanocrystalline 2D materials such as 2D magnetism, spin transport, and sensor devices[9,10,41].

## Methods

**Synthesis of NG films**. A piece of Pt foil (99.95 wt%, 150 μm thickness) was finely polished and annealed at 700 °C in air for 1 h before the first use. To polish the Pt foils, the diamond polishing agent with abrasive particle size of 3.5 μm was first used to eliminate the rough scratches caused by mechanical processing and the polishing time was about 10 min. Then, the diamond polishing agent with abrasive particle size of 0.25 μm was used to continue to polish the Pt foil for another 20 min. The surface roughness of the polished Pt foil is around 9.78 nm (Supplementary Fig. 12). After heating to the set-up temperature (800 °C, 850 °C, 900 °C, 950 °C, 1000 °C, or 1050 °C) in argon atmosphere, the Pt foil was rapidly quenched in ethanol at room temperature to grow NG films. The whole quenching process lasted a few seconds (Supplementary Movie 1). In order to demonstrate the ultrafast synthesis (Supplementary Movie 1), we used an inductive heater (HFP-25, Rijia, China) to heat the Pt foil to the set-up temperature, in which the temperature of the Pt foil was monitored using an infrared thermometer (Xi'an Henghaida Electronic Technology Co., Ltd; accuracy, ± 1 °C; response time, 10 ms) and precisely controlled through a feedback control system. The conventional tube furnace is also applicable to heating the Pt foil. If not specified, the Pt foil used was 150 μm thick.

**Transfer of NG films**. The NG films were transferred onto a 290 nm-thick SiO₂/Si substrate or TEM grid by the electrochemical bubbling method[14]. PMMA (Mw = 960,000 Da, 4 wt% in ethyl lactate) was first spin-coated on the surface of NG at a rate of 2000 r.p.m. for 1 min and then baked at 180 °C for 25 min. After that, the PMMA-coated NG/Pt foil was dipped into a NaOH (1 M) aqueous solution and was used as a cathode under a constant current of 0.2 A. After the PMMA-coated NG film was separated from the Pt substrate, it was stamped onto a SiO₂/Si substrate or TEM grid and finally PMMA was removed by hot acetone at 50 °C.

**Structural characterizations**. The morphology of NG films on Pt foil and SiO₂/Si were characterized by SEM (Nova Nano SEM 430, acceleration voltage of 5 kV) and optical microscope (Nikon LV100D), respectively. AFM (Multimode 8, Bruker) was used to characterize the thickness of NG films transferred onto the SiO₂/Si substrate in a tapping mode. XPS (ESCALAB 250 using Al Kα radiation source) was used to identify the chemical composition of NG films. The optical transmittance of NG films on PET (Polyethylene terephthalate) was measured by UV–vis-NIR spectrometer (Agilent Model Cary 5E) with wavelength from 300 to 800 nm. Raman spectra were recorded with a Raman spectrometer (JY HR800, 532 nm laser wavelength, 1 μm spot size). The laser power was below 2 mW to avoid laser-heating-induced damage on the sample. The ED pattern of NG films was recorded by TEM (FEI Tecnai T12, 120 kV). The atomic structure of NG films was characterized by TEM (FEI Titan Cube Themis G2 300 equipped with double spherical aberration correctors and a monochromator, 80 kV).

**Electrical property measurements**. The sheet resistances of the NG films on SiO₂/Si substrate were measured by a four-probe station (RTS-9) at room temperature[15]. To fabricate Hall device, the NG films on 290 nm-thick SiO₂/Si substrates were first characterized and located by an optical microscope, so that the typical regions were optically positioned relative to predefined marks. After that, we used standard electronic beam lithography (EBL) to precisely define the etching area and the unwanted NG region was removed by inductively coupled plasma to obtain standard Hall bars. Finally, electrodes consisting of Ti/Au (5/90 nm) were formed directly on the top of NG by electron beam evaporation after EBL. The width of the Hall bar device was 3 μm and the length of the channel for the four-terminal measurement was 5.5 μm. The transport measurements were performed in a Quantum Design DynaCool Physical Properties Measurement System. The resistance was measured in a four-terminal configuration using the standard low-frequency lock-in technique with a small applied AC current of 5–50 nA or voltage (<100 μV).

**Mechanical property measurements**. The Young's modulus and fracture strength of NG films were measured by AFM nanoindentation[8,26]. The NG films were first transferred on a SiO₂/Si substrate with an array of holes (1.2 μm diameter, 300 nm depth) before test. The force curves were then measured by AFM (Multimode 8, Bruker) with a diamond tip (AD-40-P60, Adama) under ambient conditions and the cantilever spring constant was 48.46 N m⁻¹, which was calibrated by the Sader method. The radius of curvature of tips was 61.58 nm. During an indentation, the sample moved up vertically with a constant rate of 50 nm s⁻¹ and the deflection of the film in z direction (Z) was obtained by subtracting the cantilever deflection from the z-piezo displacement. Finally, to acquire the fracture strength of NG film, the film was indented to failure. To measure the mechanical properties of single-crystal graphene domains (~1 mm) and the 220 nm-grained polycrystalline

graphene films, a small diamond tip (AD-150-NM, Adama) with a radius of 11.17 nm and spring constant of 180.5 N m$^{-1}$ was used.

**Simulation of the cooling process by COMSOL.** The temperature change of Pt foil as a function of cooling time was simulated by COMSOL. In the simulations, the properties of Pt foil and cooling mediums (including air and ethanol) were obtained from the database of COMSOL. The model geometry of Pt foil was defined as a cuboid (length: 10 mm; width: 10 mm), and the thickness and initial temperature of the model system were adjusted based on the real thickness and onset temperature of the Pt foil we used. When the hot Pt foil cools in air naturally, the surface radiation cooling of Pt foil and the convective cooling of air were taken into consideration. The surface emissivity of Pt foil and the convection heat transfer coefficient of air were set as 0.17 and 25 W m$^{-2}$ K$^{-1}$, respectively. When the hot Pt foil was quenched in ethanol, the surface radiation cooling of Pt foil and the two-phase flow of ethanol were taken into consideration. The surface emissivity of Pt foil was set as 0.17 and the convection heat transfer coefficients of ethanol were derived from Nusselt numbers.

## Data availability

The data that support the findings of this study are available from the corresponding author upon request.

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

## Acknowledgements

We thank Maolin Chen for the kind help on electrical property measurements, and Lichang Yin and Kangyu Zhang for drawing the atomic structure model of NG film. This work was financially supported by the National Key R&D Program of China (Number 2016YFA0200101), National Science Foundation of China (Numbers 51325205, 51290273, and 51521091), the Strategic Priority Research Program of Chinese Academy of Sciences (Number XDB30000000), Chinese Academy of Sciences (Number ZDBS-LY-JSC027), LiaoNing Revitalization Talents Program (Number XLYC1808013), SYNL-T.S. K Research Fellowship, the Youth Innovation Promotion Association of the Chinese Academy of Sciences, the Program for Guangdong Introducing Innovative and Enter-preneurial Teams (Number 2017ZT07C341), the Bureau of Industry and Information Technology of Shenzhen for the "2017 Graphene Manufacturing Innovation Center Project" (Number 201901171523), and the Development and Reform Commission of Shenzhen Municipality for the development of the "Low-Dimensional Materials and Devices" discipline.

## Author contributions

W.R. conceived and supervised the project. W.R., C.X., and T.Z. designed the growth experiments. T.Z. carried out growth experiments, Raman characterizations, and sheet resistance measurements. W.M. performed mechanical property measurements and analyzed the data. Z.B.L. performed TEM measurements. T.Y.Z. helped with growth experiments and performed quenching simulations. Z.L. carried out transport measurements under the supervision of N.K. S.F. helped with the transport measurements under the supervision of D.S. M.Z. helped with the analyses on the transport data. T.Z. and W.R. analyzed the data with the help of other authors. T.Z., H.C., and W.R. wrote the manuscript. All the authors discussed the results and commented on the manuscript.

## Competing interests

The authors declare no competing interests.
