## [Peer Review File · Nature Communications]

Reviewers' comments:

Reviewer #1 (Remarks to the Author):

In this manuscript, authors report the ultrafast growth of a nanocrystalline graphene (NG) via quenching in a low-temperature liquid carbon precursor. They show that such proposed method could synthesis uniform and scalable NG sheet and foam which exhibit good mechanical properties and semiconducting behavior.

While I found the overall work to be interesting, the manuscript can be considered after addressing several major issues as stated below:

- 1) Only two temperatures (grain size) were examined to scale both mechanical properties and electrical resistance as a function of grain size. It would be helpful to synthesis NG in various temperatures between or out of current range (900-1050 °C).
- 2) Authors need to provide elaborated details with regards to the sudden decrease (55%) of mechanical strength for the grain size below 12 nm. To me it can be an indenter tip radius effect rather than the grain size effect, since the stresses under the tip is maximum (Int. J. Non-Linear Mechanics, Vol. 3. pp. 307-324) and for the grain size below the tip radius, the measured mechanical strength is fracture strength of individual grain boundary instead of overall strength of NG. This may be cleared by larger indenter tip to check if a sudden decrease can be observed in the 10nm NG.
- 3) Coloring Fig. 1d misses useful grain boundary information. It would be helpful to provide a clear HRTEM from grain boundary.
- 4) Please provide a reference for the sentence, "The number of GB junctions is roughly scaled with the grain size in an inverse quadratic relation".
- 5) How did the authors measure the temperature and how about the temperature accuracy?

Reviewer #2 (Remarks to the Author):

The paper presents a novel method of creating a high quality nanostructured graphene with multiple grain boundaries using heating and quenching in the liquid carbon source (ethanol). Given the simplicity of the method and re-usability of the substrate, as well as quality of the resulting film, the manuscript should present an interest to the wider community and may be considered for the publication in the NatureComms once the questions below are properly addressed.

1. As the quenching process critically depends on the temperature and the time profile of the temperature on the substrate, the time profile of the cooling of the Pt films in air AND in the ethanol has to be estimated. What was the precision of the T measurements? What happens if the T is above 1050 deg C or below 900 C?
2. Also, the dependence of the growth on the thickness of the Pt film is only mentioned, requiring much more detailed analysis and reports on the bilayer growth (the bilayer nature also should be confirmed and explained in Fig. S4).
3. The quenching in ethanol may include oxidation of graphene turning it to GO. What was the oxygen content in the resulting film. That may also be a key to the drastic change in the film resistance, rather that GB explanation by the authors.
4. Fig 2e and 3e TEM images are very similar, whereas the properties of the film are VERY different, that needs appropriate explanation.
5. Mechanical properties - the explanation why the GB have minimal effect on the Young's moduli

of the film.

6. What was the doping of the film? For the film on the SiO₂/Si the Dirac point measurements should be presented, and compared between the different processing conditions.

Minor comments.

- Raman peaks in Fig.2 and 3, and S4 should be labeled.
- line 183. "blind" is not correct word in the context.
- lines 190 onwards - what was the fitting parameters for pre-stress?
- How the Pt polishing was performed? What was the resulting RMS roughness of Pt?
- what IR thermometer was used for the monitoring the T of the foil?
- Fig.5 Arrhenius plots. Specify explicitly the T range.

Reviewer #3 (Remarks to the Author):

In this article, the authors present a new straightforward procedure for producing nanocrystalline graphene (NG) films. In this procedure a metal substrate foil is heated up to high temperatures and afterwards submerged into a carbon precursor liquid. During submerging the precursor liquid decomposes resulting in high nucleation density. Through adjustments of the temperature the grain size of the forming graphene film can be modified. Mainly two films were investigated, with 12 nm and 4 nm grain sizes. As characterization for the NG films HRTEM, AFM, Raman and UV-Vis-NIR spectroscopy. Here, the authors proved the formation of monolayer films and observed a high density of grain boundaries. Afterwards, the authors compared the mechanical and electrical properties of their two NG films with single crystal graphene and 220 nm grained polycrystalline graphene. For the mechanical properties it was observed that with decreasing grain size the mechanical stability decreased as well since the defected areas of the grain boundaries would fracture first. Concerning electrical properties these also decrease with a higher amount of grain boundaries to the point where the NG films observe semiconducting behavior. Both properties derive as grain size gets lower.

This article is well written and presents a promising, although highly defect inducing alternative to CVD grown graphene.

Some points however are not thoroughly investigated. One point is the "statistical" measurements for the determination of the grain size of the NG films. 40 grains are not enough to make a statement over the average grain size of a ~1 x 1.5 cm film. Here more data would help convince the reader that the whole film experiences grain sizes of a certain size.

The other point is the Raman spectroscopy. For the NG film produced with an onset temperature of 900 °C an increase in defect concentration and a broadening of the G- and D-band is observed. Here a more in depth explanation for the increase in defect density is needed, since the question arises if this results from additional grain

Response to reviewers' comments

Reviewer #1 (Remarks to the Author):

In this manuscript, authors report the ultrafast growth of a nanocrystalline graphene (NG) via quenching in a low-temperature liquid carbon precursor. They show that such proposed method could synthesis uniform and scalable NG sheet and foam which exhibit good mechanical properties and semiconducting behavior.

While I found the overall work to be interesting, the manuscript can be considered after addressing several major issues as stated below:

Reply: We thank the reviewer very much for positive comment.

1) Only two temperatures (grain size) were examined to scale both mechanical properties and electrical resistance as a function of grain size. It would be helpful to synthesis NG in various temperatures between or out of current range (900-1050 °C).

Reply: We thank the reviewer very much for kind suggestion.

According to the reviewer's suggestion, we have synthesized graphene using Pt foil with 5 additional onset temperatures (800 °C, 850 °C, 950 °C, 1000 °C, and 1100 °C) by the ethanol quenching method. As shown in Fig. R1, only discontinuous graphene films can be synthesized at 850 °C and 800 °C, while a great number of adlayers appear at 1100 °C. Therefore, we only characterized the graphene synthesized at 950 °C and 1000 °C by using aberration-corrected high-resolution transmission electron microscopy (HRTEM). Here, 120 and 102 grains were measured for the former and latter samples, respectively. The results show that both samples are NG films, which have average grain size (L_g) of ~5.8 nm and ~8.0 nm, respectively (Fig. R2). To give a more accurate evaluation on the grain size, we also measured more grains for the NG films synthesized at 900 °C (108 grains) and 1050 °C (111 grains), and the obtained average grain size is ~3.6 nm and ~10.3 nm, respectively (Fig. R2).

We then measured the electrical properties of the two new NG films using a four-probe station at room temperature. The obtained sheet resistance of 5.8-nm- and 8.0-nm-grained NG films is ~80 k Ω sq⁻¹ and 49 k Ω sq⁻¹, respectively. It can be seen that the increase in sheet resistance is speeded up dramatically with decreasing the grain size at the nanometer scale (Fig. R3a). We also plotted these two new data together with the previous 7 data. Similar to the original results, no good fit can be obtained for all the data, but an excellent fit was achieved when the data for 3.6-nm- and 5.8-nm-grained NG was excluded (Fig. R3b). The unexpectedly large resistances of 3.6-nm- and 5.8-nm-grained NG indicate strongly enhanced influence of the GBs on the electronic properties at nanometer scale.

As the reviewer mentioned in the second comment, when the tip radius is very small, the measured mechanical strength might be the fracture strength of individual grain boundary (GB) instead of overall strength of NG. Therefore, we used a larger indenter tip (61.58 nm radius) to measure the mechanical properties of the 4 kinds of

NG films. For the 3.6-nm-grained NG film, the extracted mean 2D Young's modulus (E^{2D}) and mean 2D fracture strength (σ^{2D}) are 193 and 33.99 N m⁻¹, respectively (Fig. R4). These values are larger than those (E^{2D} : 170.2 N m⁻¹; σ^{2D} : 25.46 N m⁻¹) measured with a small diamond tip (11.17 nm radius), which can be attributed to the indenter tip radius effect mentioned by the reviewer. For the 10.3-nm-grained NG film, however, the mean E^{2D} and σ^{2D} measured with 61.58-nm diamond tip (298.7 N m⁻¹ and 42.54 N m⁻¹, Fig. R4) are comparable to the previous results (E^{2D} : 297.8 N m⁻¹; σ^{2D} : 44.89 N m⁻¹), indicating that the indenter tip radius effect is negligible for the NG film with large grains. These results suggest that the use of a large indenter tip is more reasonable to obtain the overall mechanical properties of NG film when the grain size is below 10 nm.

Figure R4 shows the E^{2D} and σ^{2D} of 4 kinds of NG films measured with 61.58-nm diamond tip. Compared to the single-crystal graphene, the E^{2D} and σ^{2D} are decreased by 9%, 17%, 28%, 43%, 46% and 9%, 25%, 36%, 39%, 40%, respectively, for the 220-nm-grained polycrystalline graphene, 10.3-nm-grained NG, 8.0-nm-grained NG, 5.8 nm-grained NG and, 3.6-nm-grained NG. Same as the previous conclusion, these results suggest that the mechanical properties of graphene reduce more pronounced but still retain high values at nanometer scale.

We have added these new data and revised the related discussions in the revised manuscript.

Figure R1. Typical optical images and Raman spectra of the graphene films synthesized with different onset temperatures. **a, b**, 800 °C. **c, d**, 850 °C. **e, f**, 1100 °C. The Pt foils used are 150 μm in thickness, and the samples have been transferred onto SiO₂/Si substrates by the bubbling method for characterizations.

Figure R2. Grain size distributions of the NG films synthesized with the onset temperature of 900 °C (a), 950 °C (b), 1000 °C (c), and 1050 °C (d), which were extracted from a great number of HRTEM images for each sample.

Figure R3. The sheet resistance as a function of grain size for the NG films only (a) and both the NG films and polycrystalline graphene films (b). The data of polycrystalline graphene films with average grain size of ~220 nm, 470 nm, 720 nm, 1 μm, and 10 μm was reproduced from our previous report [Nat. Commun., 8, 14486 (2017)]. The error bars show the variations of sheet resistance. The blue line is the fitting curve to the equation $R_s = R_s^G + \rho_{GB}/L_g$.

Figure R4. Representative force-displacement curves (a) and histograms of elastic stiffness (b) and fracture strength (c) of 3.6-nm-, 5.8-nm-, 8.0-nm- and 10.3-nm-grained NG films. The red lines in a are the fitting curves, and dashed lines in b and c represent Gaussian fits to the data.

2) Authors need to provide elaborated details with regards to the sudden decrease (55%) of mechanical strength for the grain size below 12 nm. To me it can be an indenter tip radius effect rather than the grain size effect, since the stresses under the tip is maximum (Int. J. Non-Linear Mechanics, Vol. 3. pp. 307-324) and for the grain size below the tip radius, the measured mechanical strength is fracture strength of individual grain boundary instead of overall strength of NG. This may be cleared by larger indenter tip to check if a sudden decrease can be observed in the 10 nm NG.

Reply: We thank the reviewer very much for the constructive comments.

According to the reviewer's suggestion, we have performed nanoindentation measurements on the 3.6-nm- and 10.3-nm-grained NG films with a larger diamond tip of 61.58 nm radius. For the 3.6-nm-grained NG films, the extracted mean 2D Young's modulus (E^{2D}) and mean 2D fracture strength (σ^{2D}) are 193 and 33.99 $N m^{-1}$, respectively. Compared with the previous results (E^{2D} : 170.2 $N m^{-1}$; σ^{2D} : 25.46 $N m^{-1}$) measured by using a small diamond tip of 11.17 nm radius, these values increase by 13.4% and 33.5%, respectively, which can be attributed to the indenter tip radius effect mentioned by the reviewer. For the 10.3-nm-grained NG film, however, the E^{2D} and σ^{2D} measured by the 61.58-nm-radius diamond tip (298.7 $N m^{-1}$ and 42.54 $N m^{-1}$) are comparable to the previous results (E^{2D} : 297.8 $N m^{-1}$; σ^{2D} : 44.89 $N m^{-1}$), indicating that the indenter tip radius effect is negligible for the NG films with large grains. These results suggest that the use of a large indenter tip is more reasonable to

obtain the real overall mechanical properties of graphene film when the grain size is below 10 nm.

To obtain the real influence of grain size on the mechanical properties of NG film, we measured the mechanical properties of all the NG films (average grain size: 3.6 nm, 5.8 nm, 8.0 nm, and 10.3 nm) using a diamond tip of 61.58 nm radius, and the results are shown in Fig. R4 and R5. As shown in Fig. R5, we still observed a sudden decrease in the mechanical strength of NG films, confirming that the sudden decrease is intrinsically attributed to the grain size effect.

We have added these new data and revised the related discussions in the revised manuscript.

Figure R5. The Young's modulus (a) and fracture strength (b) of graphene films as a function of grain size. The inset in b is a fit (blue curve) to the experimental data, which gives a fitted equation $\sigma = 93.5 \times L_g^{0.09}$.

3) Coloring Fig. 1d misses useful grain boundary information. It would be helpful to provide a clear HRTEM from grain boundary.

Reply: We thank the reviewer very much for kind reminder. We have removed the colored overlayer to clearly show the grain boundaries.

4) Please provide a reference for the sentence, "The number of GB junctions is roughly scaled with the grain size in an inverse quadratic relation".

Reply: According to a previous report [Scientific Reports, 4: 5991 (2014)], the density of GB junction (number of GB junctions per nm², ρ_{GB}) is scaled as a function of the average grain size (L_g), $\rho_{GB} \propto L_g^{-2}$. We have cited this paper in our manuscript.

5) How did the authors measure the temperature and how about the temperature accuracy?

Reply: The temperature of the Pt foil was monitored using an infrared thermometer and controlled through a feedback control system. The accuracy of infrared thermometer is ± 1 °C and the response time is 10 ms. Such high accuracy and rapid response allow us to accurately measure the temperature of the Pt foil.

We have added this information in the revised manuscript.

Reviewer #2 (Remarks to the Author):

The paper presents a novel method of creating a high quality nanostructured graphene with multiple grain boundaries using heating and quenching in the liquid carbon source (ethanol). Given the simplicity of the method and re-usability of the substrate, as well as quality of the resulting film, the manuscript should present an interest to the wider community and may be considered for the publication in the Nature Comms once the questions below are properly addressed.

Reply: We thank the reviewer very much for positive comment.

1. As the quenching process critically depends on the temperature and the time profile of the temperature on the substrate, the time profile of the cooling of the Pt films in air AND in the ethanol has to be estimated. What was the precision of the T measurements? What happens if the T is above 1050 deg C or below 900 deg C?

Reply: We thank the reviewer very much for valuable suggestion.

The temperature of the Pt foil was monitored using an infrared thermometer and controlled through a feedback control system. The accuracy of infrared thermometer is ± 1 °C. However, due to the fast cooling rate of hot Pt foil in air and ethanol, it is difficult to accurately measure the time profile of the cooling of the Pt foil directly. Instead, we simulated the temperature change of Pt foil as a function of cooling time by COMSOL, which has been widely used to simulate the quenching process of metals in various quenching mediums. In the simulations, the properties of Pt foil and cooling mediums (including air and ethanol) were obtained from the database of COMSOL. Because it takes about 0.3 s to move the Pt foil from the heater coil into ethanol, we chose the period of 1 s to analyze the cooling process of Pt foil in air.

Figure R6a and R6b show the simulation results in air and ethanol, respectively. It can be seen that the cooling rate of Pt foil decreases with decreasing the onset temperature in both air and ethanol, which is due to the less heat storage in a lower temperature Pt. For the onset temperature of 900, 950, 1000 and 1050 °C, the temperature of Pt foil decreases to 847, 892, 935 and 979 °C, respectively, before immersing into ethanol (Table R1), which are the real onset temperature of Pt foil during the quenching process. In particular, the quenching time, during which the temperature of Pt foil decreases from the real onset temperature to the decomposition temperature of ethanol, decreases with decreasing the onset temperature. The lower onset temperature and shorter quenching time lead to decreased carbon feeding.

We have studied the synthesis of graphene by quenching Pt foils with different onset temperatures of 800, 850, or 1100 °C into ethanol at room temperature. As shown in Fig. R1, only discontinuous graphene is obtained with the onset temperatures of 800 °C and 850 °C. This is due to the insufficient carbon feeding discussed above. In contrast, because of the excessive carbon feeding at high onset temperature of 1100 °C, a great number of adlayers are formed on the surface of monolayer NG film.

These new data and related discussions have been added in the revised manuscript.

Figure R6. The temperature profiles of 150- μm -thick Pt foils with different onset temperatures during cooling in air (a) and subsequently cooling in ethanol (b).

Table R1. The real onset temperatures of 150- μm -thick and 1-mm-thick Pt foils with different onset temperatures during the ethanol quenching process, which are cooled in air for ~ 0.3 s before immersing into ethanol.

Thickness (mm)	Onset temperature (°C)	Real onset temperature (°C)
0.15	800	757
0.15	850	803
0.15	900	847
0.15	950	892
0.15	1000	935
0.15	1050	979
0.15	1100	1022
1.0	900	889
1.0	1050	1036

2. Also, the dependence of the growth on the thickness of the Pt film is only mentioned, requiring much more detailed analysis and reports on the bilayer growth (the bilayer nature also should be confirmed and explained in Fig. S4).

Reply: We thank the reviewer very much for constructive suggestion.

To understand the influence of the thickness of Pt foil on graphene growth, the cooling processes of 1-mm-thick Pt foil in both air and ethanol were simulated by COMSOL. As shown in Fig. R7, the cooling rate of 1-mm-thick Pt foil is much slower than that of 150- μm -thick Pt foil because thicker Pt can store more heat. For the onset temperature of 900 °C and 1050 °C, the real onset temperature of Pt foil during the quenching process is 889 °C and 1036 °C, respectively (Fig. R7a and Table R1). More importantly, the quenching time of 1-mm-thick Pt foil in ethanol is

significantly longer than that of 150- μm -thick Pt foil (Fig. R7b). Therefore, much more carbon species are supplied during the quenching of 1-mm-thick Pt foil, leading to the growth of thicker graphene film.

We further characterized the graphene grown on 1-mm-thick Pt foil with onset temperature of 1050 $^{\circ}\text{C}$ by using atomic force microscopy (AFM). It is found that the graphene is dominantly bilayer covered with some islands of $\sim 2\ \mu\text{m}$ (Fig. R8). As a result, the whole film shows an optical transmittance of 94.6% at 550 nm, which is a little bit lower than that of perfect bilayer (95.4%).

We have added the above new data and revised the related discussions in the revised manuscript.

Figure R7. The temperature profiles of Pt foils with different thicknesses and onset temperatures during cooling in air (a) and subsequent cooling in ethanol for onset temperature of 1050 $^{\circ}\text{C}$ (b).

Figure R8. AFM images of the graphene film synthesized by quenching a 1-mm-thick Pt foil with the onset temperature of 1050 $^{\circ}\text{C}$ in ethanol.

3. The quenching in ethanol may include oxidation of graphene turning it to GO. What was the oxygen content in the resulting film. That may also be a key to the drastic change in the film resistance, rather than GB explanation by the authors.

Reply: We thank the reviewer very much for kind comment.

X-ray photoelectron spectroscopy (XPS) has been extensively used to characterize the oxidation degree of graphene oxide (GO). Typically, the XPS C1s spectra of GO show strong C=C peak (284.6 eV), prominent epoxy/hydroxyl peak (C-O, 287.0 eV), and weak carbonyl (C=O, 288.0 eV) and carboxyl (O-C=O, 289.2 eV) peaks. To identify whether the NG films are oxidized, we used XPS to characterize the as-synthesized 3.6-nm-grained and 10.3-nm-grained NG films on Pt foils. For comparison, polycrystalline graphene films with grain size of ~500 μm , which were synthesized by chemical vapour deposition (CVD) on Pt foils using methane as carbon precursor, were also measured. Note that all these samples show only a single C1s peak at 284.6 eV (Fig. R9), corresponding to the graphite-like sp^2 -hybridized carbon. These results confirm that the NG films are free of oxidation even though oxygen-containing ethanol is used as carbon source.

We have added these new data and related discussions in the revised manuscript.

Figure R9. XPS C1s spectra of as-synthesized 3.6-nm-grained NG (a), 10.3-nm-grained NG (b), and 500- μm -grained polycrystalline graphene film grown by conventional CVD with methane as carbon precursor (c).

4. Fig 2e and 3e TEM images are very similar, whereas the properties of the film are VERY different, that needs appropriate explanation.

Reply: We thank the reviewer very much for kind comment.

The TEM images shown in original Fig. 2e and 3e indeed looks similar, but the grain size has big difference. For comparison, we put these two images together as shown in Fig. R10. It can be seen that the grain size of each two grains indicated by yellow arrows in Fig. R10a and R10b are ~10 nm and ~4 nm, respectively. We have measured 111 and 108 grains for the former and latter samples, respectively, and the obtained average grain size is 10.3 nm and 3.6 nm (Fig. R2). According to previous report [Carbon 96, 429 (2016)], the concentration of carbon atoms in the grain boundary regions is ~5% for $L_g = 10.3$ nm, while it is ~15% for $L_g = 3.6$ nm. Intrinsically, grain boundary is a kind of topological defect. The big difference in grain size and defect concentration leads to very different mechanical and electrical properties for these two samples.

Figure R10. Typical HRTEM images of 10.3-nm-grained NG (a) and 3.6-nm-grained NG (b). Inset, Fourier transform patterns. To expose the GBs, the samples were annealed at 320 °C in air for 8 h to remove the adsorbed PMMA residues, which simultaneously generates some holes indicated by red arrows.

5. Mechanical properties - the explanation why the GB have minimal effect on the Young's moduli of the film.

Reply: We thank the reviewer very much for kind suggestion.

As shown in Fig. R5, GB actually has a similar effect on Young's modulus and fracture strength, both of which reduce more pronounced at nanometer scale. Compared to the single-crystal graphene, the mean 2D Young's modulus (E^{2D}) and mean 2D fracture strength (σ^{2D}) are decreased by 9%, 17%, 28%, 43%, 46% and 9%, 25%, 36%, 39%, 40%, respectively, for the 220-nm-grained polycrystalline graphene, 10.3-nm-grained NG, 8.0-nm-grained NG, 5.8 nm-grained NG and, 3.6-nm-grained NG. In the revised manuscript, we have added more discussions about the influence of grain size on the Young's modulus of NG films as shown below.

For nanocrystalline metals, a reduction in grain size only leads to a slight decrease in elastic constant compared to the corresponding polycrystals. [Nature 391, 561-563 (1998); J. Mater. Res. 10, 2892-2896 (2011); Mater. Sci. Eng. A 234-236, 77-82 (1997)]. As for NG films, GB is a low-elastic modulus component because the bonds in GBs are usually longer than those in the grains [Nano Lett. **13**, 1829-1833 (2013); Mater. Sci. Eng. B 198, 95-101 (2015)]. Compared to bulk nanocrystalline metals, NG films show more pronounced decrease in Young's modulus with decreasing grain size, indicating more significant effect of GB on the Young's modulus in this atomically thin system.

6. What was the doping of the film? For the film on the SiO₂/Si the Dirac point measurements should be presented, and compared between the different processing conditions.

Reply: We thank the reviewer very much for kind suggestion.

Figure R11a and R11b show the room temperature field effect characteristics of 10.3-nm-grained and 3.6-nm-grained NG films, respectively. The Dirac point of both NG films is located in the positive gate voltage region, indicating a strong hole-doping effect. With decreasing the grain size of NG films, the Dirac point shifts toward a more positive gate voltage. It is well known that the presence of adsorbed molecules and polymer residue on the surface of the graphene can induce strong doping in graphene [Nano Lett. 13 1462 (2013); Nano Lett. 10, 1149 (2010)]. As shown in the main text (Fig. 2e and 3e), the PMMA residues tend to adsorb on the GBs in NG films to form a network because of the stronger adsorption ability of GBs than the perfect graphene lattice. As reported previously, PMMA always results in strong hole doping in graphene devices [Appl. Phys. Lett. 99, 122108 (2011)]. Therefore, the observed doping behavior can be attributed to the PMMA residues on the surface of NG films. With a decrease in grain size, more PMMA residues are adsorbed, and as a result, the Dirac point shifts toward a larger positive value in gate voltage (Figure 11Rb).

We have added these data and related discussions in the revised manuscript.

Figure R11. Four terminal resistance as a function of back gate voltage of representative 10.3-nm-grained (a) and 3.6-nm-grained (b) NG films, measured at room temperature. The Dirac point of both NG samples is located in the positive gate voltage region, demonstrating hole doping of the graphene. The Dirac point shifts toward a more positive gate voltage with decreasing grain size.

Minor comments.

- Raman peaks in Fig.2 and 3, and S4 should be labeled.

Reply: We have labeled the D, G, D' and 2D peaks in these figures.

- line 183. "blind" is not correct word in the context.

Reply: We have changed “blind” to “blank”.

- lines 190 onwards - what was the fitting parameters for pre-stress?

Reply: The “ σ_0^{2D} ” in Eq. 1 is the fitting parameter for the pre-stress. We have added and highlighted it in the revised manuscript.

- How the Pt polishing was performed? What was the resulting RMS roughness of Pt?

Reply: To polish the Pt foils, two kinds of polishing agents (diamond, Shenyang Kejing Auto-Instrument Co., Ltd, abrasive particle size of 3.5 μm and 0.25 μm) were used. First, the polishing agent with abrasive particle size of 3.5 μm was used to eliminate the rough scratches caused by mechanical processing and the polishing time was about 10 min. After that, the polishing agent with abrasive particle size of 0.25 μm was used to continue to polish the Pt foil for another 20 min. As shown in Fig. R12a and R12b, the rough scratches disappear and only fine scratches can be observed after polishing. AFM measurements show that the RMS roughness of the polished Pt surface is 9.78 nm.

We have added these new data and information in the revised manuscript.

Figure 12. Typical surface topography of Pt foil after polishing. **a**, Optical image. **b**, SEM image. **c**, AFM image.

- what IR thermometer was used for the monitoring the T of the foil?

Reply: The infrared thermometer used in our study was purchased from Xi'an Henghaida Electronic Technology Co., Ltd. Its accuracy is ± 1 °C and response time is 10 ms. We have added this information in the revised manuscript.

- Fig.5 Arrhenius plots. Specify explicitly the T range.

Reply: The T range is 30 – 300 K for 10.3-nm-grained NG and 150 – 300 K for 3.6-nm-grained NG. We have added this information in the revised manuscript.

Reviewer #3 (Remarks to the Author):

In this article, the authors present a new straightforward procedure for producing nanocrystalline graphene (NG) films. In this procedure a metal substrate foil is heated up to high temperatures and afterwards submerged into a carbon precursor liquid. During submerging the precursor liquid decomposes resulting in high nucleation density. Through adjustments of the temperature the grain size of the forming graphene film can be modified. Mainly two films were investigated, with 12 nm and 4 nm grain sizes. As characterization for the NG films HRTEM, AFM, Raman and UV-Vis-NIR spectroscopy. Here, the authors proved the formation of monolayer films and observed a high density of grain boundaries. Afterwards, the authors compared the mechanical and electrical properties of their two NG films with single crystal graphene and 220 nm grained polycrystalline graphene. For the mechanical properties it was observed that with decreasing grain size the mechanical stability decreased as well since the defected areas of the grain boundaries would fracture first. Concerning electrical properties these also decrease with a higher amount of grain boundaries to the point where the NG films observe semiconducting behavior. Both properties derive as grain size gets lower.

This article is well written and presents a promising, although highly defect inducing alternative to CVD grown graphene.

Reply: We thank the reviewer very much for positive comment.

Some points however are not thoroughly investigated.

One point is the “statistical” measurements for the determination of the grain size of the NG films. 40 grains are not enough to make a statement over the average grain size of a ~1 x 1.5 cm film. Here more data would help convince the reader that the whole film experiences grain sizes of a certain size.

Reply: We thank the reviewer very much for kind suggestion.

We have measured more grains by using aberration-corrected high-resolution transmission electron microscopy (HRTEM) to give statistical information of the grain size (L_g) of all the NG samples. 108, 120, 102, and 111 grains were measured for the NG samples synthesized with onset temperature of 900, 950, 1000 and 1050 °C, respectively, and the obtained average grain size is 3.6, 5.8, 8.0, and 10.3 nm (Fig. R13). Due to the strong adsorption ability of grain boundaries (GBs), it is hard to get atomic image of NG. To expose the whole grain and GBs, the samples were annealed at 320 °C in air for 8 h to remove the adsorbed PMMA residues, which make only a few grains exposed in the vision area. Each statistical result shown in Fig. R13 was actually obtained by measuring many different samples synthesized under the same condition and different areas of the same sample, which therefore represents the overall grain size information of each sample.

We have added these new data and changed the related discussions in the revised manuscript.

Figure R13. Grain size distributions of the NG films synthesized with the onset temperature of 900 °C (a), 950 °C (b), 1000 °C (c), and 1050 °C (d), which were extracted from a great number of HRTEM images for each sample.

The other point is the Raman spectroscopy. For the NG film produced with an onset temperature of 900 °C an increase in defect concentration and a broadening of the G- and D-band is observed. Here a more in depth explanation for the increase in defect density is needed, since the question arises if this results from additional grain.

Reply: We thank the reviewer very much for kind suggestion.

The grain boundary (GB) in graphene is a topological line defect formed by chains of pentagons, heptagons and distorted hexagons [Nature 469, 389-392 (2011)]. Based on our HRTEM observations, the interior of the grains has very high crystallinity for all the NG samples. Therefore, the defects in our NG films are dominantly GBs, and the dramatic change observed in Raman spectra of 900 °C sample is due to the significantly increased GB density. According to previous report [Carbon 95, 646 (2015)], if $L_g < l_C$, the Raman-allowed phonon wavevector q is relaxed due to the spatial confinement of the crystallites, leading to the broadening of the Raman peaks, where L_g is the grain size, and l_C refers to the coherence length of optical phonons (~30 nm). Moreover, the width of both G peak and D peak increases exponentially as L_g decreases to 0 [Carbon 95, 646 (2015); Phys. Rev. B 82, 125429 (2010)]. Therefore, greatly broadened D and G peaks are observed in the NG films produced with an onset temperature of 900 °C, i.e., 3.6-nm-grained NG.

We have added the above discussions in the revised manuscript.

REVIEWERS' COMMENTS:

Reviewer #1 (Remarks to the Author):

The work entitled "Quenching growth of nanocrystalline graphene films and grain size dependent strength and bandgap opening" by Prof Ren and colleagues has been highly improved and addressed the reviewer's comments. It is acceptable for publication in Nature Communications.

Reviewer #2 (Remarks to the Author):

The authors provided clear answers to questions of all reviewers, and included relevant data in the manuscript. The paper as it stands presents an interesting topic that is of interest to a wide community, and can now be recommended for the publication in Nature Communications.

Reviewer #3 (Remarks to the Author):

The authors report on a novel and easy-operation method for the preparation of a high quality nanostructured graphene containing multiple grain boundaries, where the effect of grain size on the mechanical and electrical properties of nanostructured graphene was deeply explored. The characterization now is convincing, the questions are also well addressed and the manuscript is well written. For these reasons, the article should be published in the Nature Comms without delay.

Response to reviewers' comments

Reviewer #1 (Remarks to the Author):

The work entitled “Quenching growth of nanocrystalline graphene films and grain size dependent strength and bandgap opening” by Prof Ren and colleagues has been highly improved and addressed the reviewer’s comments. It is acceptable for publication in Nature Communications.

Reply: We thank the reviewer very much for positive comments.

Reviewer #2 (Remarks to the Author):

The authors provided clear answers to questions of all reviewers, and included relevant data in the manuscript. The paper as it stands presents an interesting topic that is of interest to a wide community, and can now be recommended for the publication in Nature Communications.

Reply: We thank the reviewer very much for positive comments.

Reviewer #3 (Remarks to the Author):

The authors report on a novel and easy-operation method for the preparation of a high quality nanostructured graphene containing multiple grain boundaries, where the effect of grain size on the mechanical and electrical properties of nanostructured graphene was deeply explored. The characterization now is convincing, the questions are also well addressed and the manuscript is well written. For these reasons, the article should be published in the Nature Comms without delay.

Reply: We thank the reviewer very much for positive comments.